# Revisiting Contrastive Divergence for Density Estimation and Sample Generation

**Azwar Abdulsalam**  *azwar.abdulsalam1729@gmail.com*
*Elmore School of Electrical and Computer Engineering*
*Purdue University*

**Joseph G. Makin**  *jgmakin@purdue.edu*
*Elmore School of Electrical and Computer Engineering*
*Purdue University*

**Reviewed on OpenReview:** *https: // openreview. net/ forum? id= i5K4SZeqtq*

## Abstract

Energy-based models (EBMs) have recently attracted renewed attention as models for complex distributions of data, like natural images. Improved image generation under the maximum-likelihood (MLE) objective has been achieved by combining very complex energy functions, in the form of deep neural networks, with Langevin dynamics for sampling from the model. However, Nijkamp and colleagues have recently shown that such EBMs become good generators without becoming good density estimators: an impractical number of Langevin steps is typically required to exit the burn-in of the Markov chain, so the training merely sculpts the energy landscape near the distribution used to initialize the chain. Careful hyperparameter choices and the use of persistent chains can significantly shorten the required number of Langevin steps, but at the price that new samples can be generated only in the vicinity of the persistent chain and not from noise. Here we introduce a simple method to achieve both convergence of the Markov chain in a practical number of Langevin steps ($L = 500$) and the ability to generate diverse, high-quality samples from noise. Under the hypothesis that Hinton's classic *contrastive-divergence* (CD) training does yield good density estimators, but simply lacks a mechanism for connecting the noise manifold to the learned data manifold, we combine CD with an MLE-like loss. We demonstrate that a simple ConvNet can be trained with this method to be good at generation as well as density estimation for CIFAR-10, Oxford Flowers, and a synthetic dataset in which the learned density can be verified visually.

## 1 Introduction

One long-standing approach to fitting probability distributions to observed data is to model the distribution only up to the normalizer. Relaxing the constraint that the model integrate to unity allows for the use of any arbitrary function from the space of data to the real line, including deep neural networks. Exponentiating this "energy" function (or, by convention, its additive inverse) yields an unnormalized probability distribution:

$$\hat{p}(\boldsymbol{x}; \boldsymbol{\theta}) \propto \exp\{-E(\boldsymbol{x}, \boldsymbol{\theta})\}. \tag{1}$$

(Circumflexes indicate models throughout.) The flexibility comes at a price, however. The classical approach to fitting distributions to data is to minimize the relative entropy of these distributions, but for unnormalized models this requires taking expectations under the model distribution. In particular, the gradient of the relative entropy is (Hinton, 1999) (see Section A.5):

$$\frac{\mathrm{d}}{\mathrm{d}\boldsymbol{\theta}} \mathrm{D}_{\mathrm{KL}}\{p(\boldsymbol{X}) \| \hat{p}(\boldsymbol{X}; \boldsymbol{\theta})\} = \mathbb{E}_{\boldsymbol{X}}\left[\frac{\partial E}{\partial \boldsymbol{\theta}}(\boldsymbol{X}, \boldsymbol{\theta})\right] - \mathbb{E}_{\hat{\boldsymbol{X}}}\left[\frac{\partial E}{\partial \boldsymbol{\theta}}(\hat{\boldsymbol{X}}, \boldsymbol{\theta})\right]. \tag{2}$$

The more complex the energy function, the less likely the integral implicit in the second term is to be analytically tractable. (The first expectation, under the data, $\boldsymbol{X}$, will be approximated by a sample average.) Low-dimensional observation spaces can be discretized and the integrals replaced with sums; but datasets of interest today, like images, occupy far too many dimensions for this approach to be feasible, since the required number of discrete bins grows exponentially in dimensionality.

The dominant approach to approximating Eq. 2 is therefore to replace the intractable expectation with an empirical average under samples drawn from the model distribution, typically with some form of Markov-chain Monte Carlo (MCMC). In MCMC, an initial, randomly chosen sample is updated according to some transition operator that guides it into the "thick" part of the distribution of interest, in this case the model represented by Eq. 1. More precisely, the samples generated by this procedure are guaranteed, after some number of iterations—the "burn-in"—to be distributed according to Eq. 1. The most powerful procedures, like Hamiltonian Monte Carlo and Langevin dynamics (LD) (Neal, 2011), exploit the gradient of the energy, $\partial E / \partial \boldsymbol{x}$, to accelerate convergence. For example, the transition operator in LD encourages descent of the energy but also encourages exploration by injecting Gaussian noise:

$$\hat{\boldsymbol{X}}_{l+1} = \hat{\boldsymbol{X}}_l - \epsilon \frac{\partial E}{\partial \boldsymbol{x}}(\hat{\boldsymbol{X}}_l, \boldsymbol{\theta}) + \sqrt{2\epsilon}\hat{\boldsymbol{Z}}_l, \tag{3}$$

where $\hat{\boldsymbol{Z}}_l \sim \mathcal{N}(\boldsymbol{0}, \boldsymbol{I})$. Continuous-time Langevin dynamics is guaranteed to have Eq. 1 as its stationary distribution, but Eq. 3, its discretization, is not (except in the limit as the step size $\epsilon$ approaches 0). A Metropolis adjustment (Metropolis et al., 1953) is therefore required in theory, although frequently neglected in practice. The discrete Langevin dynamics of Eq. 3 can also be used to generate samples from trained models at test time, as well as to evaluate the loss gradient, Eq. 2, during training.

Recently, energy-based models (EBMs) trained by descending the gradient of Eq. 2 have performed well at conditional and unconditional image generation, generating molecular graphs, image completion, compositional generation, and out-of-distribution generalization (Nijkamp et al., 2019; 2020; Du & Mordatch, 2019; Du et al., 2021; 2023; Liu et al., 2021; Gao et al., 2021). Recent work has demonstrated that EBMs trained and tested in this or similar ways can generate realistic images in a variety of contexts (Zhao et al., 2021; Xiao et al., 2021a; Xie et al., 2022; Pang et al., 2020), e.g. by using a ConvNet as the energy function. And yet, contrary to the orthodox theory just reviewed here (and underlying the papers just cited), it seems that none of these EBMs has learned the data distribution (Nijkamp et al., 2019; 2020). More precisely, the distribution generated by the sampling process in Eq. 3 *does* resemble the data distribution, even though Eq. 1 does not. In an important pair of papers, Nijkamp and colleagues showed that for the parameter choices typically employed in EBM training, in particular for any number of Langevin steps $L$ less than about 10,000 ($L = 100$ is typical), the Markov chain does not make it out of the burn-in stage. Consequently, the training sculpts this energy landscape instead of that of the model.

Thus the training procedure with $L \approx 100$, which Nijkamp et al. (2019; 2020) call "non-convergent, short-run MCMC," yields a useful generator. But the resulting EBM cannot be used to assign probabilities to samples, either relative or absolute, nor consequently to identify out-of-distribution data. Nijkamp and colleagues argue that the Langevin dynamics can be adjusted to yield samples from the model distribution, i.e. to get past the burn-in, with careful choice of the step size, $\epsilon$; and with "persistent initialization" of the Markov chain, i.e. initializing $\hat{\boldsymbol{X}}_1$ at training iteration $i$ with $\hat{\boldsymbol{X}}_L$ from training iteration $i-1$. ($\hat{\boldsymbol{X}}_1$ at training iteration $i = 1$ is drawn from a noise distribution.) But this makes generation of new data dependent upon a bank of persistent samples. In addition to imposing a storage cost (or a computational cost, if the bank is to be regenerated from scratch), any finite sample bank also limits the diversity of new samples.

In sum, EBMs putatively trained by descent of the gradient in Eq. 2—or, equivalently, to maximize the likelihood of the model parameters under the observed data—either fail to assign the correct probabilities to data, cannot generate new samples from noise, or require impractically long training ($L > 10,000$).

Here we propose and verify a simple procedure that satisfies all three desiderata. Our point of departure is the observation that density estimation and generation make demands on the model that are to some extent orthogonal. Assigning the right probabilities to data requires getting the energy function correct on or near the data (e.g., image) manifold, whereas generating new samples from noise requires that the energy steadily decrease from the "noise manifold" (e.g., for high-dimensional Gaussian noise, a hypersphere)

toward the data manifold. These two manifolds will not in general intersect, so these are distinct demands. Nevertheless, there exist losses that plausibly enforce each of them. *Short-run MCMC* evidently sculpts a path from noise manifold to data manifold. And we hypothesize, and then show, that Hinton's method of *contrastive divergence* (CD) is, despite recent work on its putative shortcomings (discussed below), in fact an excellent density estimator. This has been overlooked because CD does not yield good generators; but on our account, this is to be expected, since it does not shape the energy landscape far from the data manifold.

In this study, we show that combining these two training methods in the right proportions yields high-quality sample generation *and* out-of-distribution detection on datasets ranging in difficulty from simple synthetic data up to CIFAR-10 and Oxford Flowers, outperforming recently proposed alternatives for training EBMs. Generation requires only a few hundred ($L \approx 500$) steps of Langevin dynamics. But unlike in models trained with short-run MCMC alone, the Langevin sampler continues to produce high-quality images from the data distribution for as long it is run (out to at least 100,000 samples).

In what follows, we elaborate on the underlying theory (Section 2), describe our methods in detail (Section 3), present these results (Section 4), and discuss their implications (Section 5).

## 2 Remedies for MLE

The problem of "non-convergent MCMC" (Nijkamp et al., 2020) has its roots in the slow rate of convergence of Langevin dynamics for non-convex functions. Under standard theory, the initial state of an ergodic Markov chain will be "forgotten" after some number of transitions—the burn-in period. However, the convergence rate is provably exponential in (a measure of) the non-convexity of the energy (Cheng et al., 2020). Energies defined by neural networks are (unless otherwise constrained) non-convex, which is consistent with the exceedingly slow convergence rates observed in practice (Nijkamp et al., 2019; 2020).

**Persistent Markov chains?** A potential remedy is to allow the Markov chain, originally initialized at a random sample, to "persist" across weight updates. The idea is that the first weight update, although occurring before the Markov chain has converged to its stationary distribution, will not alter that stationary distribution so much as to render the samples up to that point useless. Intuitively, if the weight updates change the stationary distribution slowly enough, the Markov chain will eventually "catch up" and enter the typical set of the model distribution, greatly decreasing the required burn-in period. In practice, multiple Markov chains are maintained simultaneously in a "bank" of samples, in order to provide an unbiased estimate of the expected energy gradient,

$$\mathbb{E}_{\hat{\boldsymbol{X}}}\left[\frac{\partial E}{\partial \boldsymbol{\theta}}(\hat{\boldsymbol{X}}, \boldsymbol{\theta})\right] \approx \left\langle \frac{\partial E}{\partial \boldsymbol{\theta}}(\hat{\boldsymbol{X}}, \boldsymbol{\theta}) \right\rangle_{\Pi_{\mathrm{p}}(\hat{\boldsymbol{X}})}. \tag{4}$$

The angle brackets indicate a Monte Carlo estimate (sample average) under the samples from the persistent Markov chains, $\Pi_{\mathrm{p}}(\hat{\boldsymbol{X}})$.

However, as we have noted, this holds generation hostage to the sample bank, whereas one would prefer to generate new images at will from noise. To alleviate this, Du & Mordatch (2019); Du et al. (2021) propose to reserve some fraction of the sample bank, typically about 5%, for noise samples. This does improve the quality of samples generated from noise, but with limited mode coverage. To increase sample diversity, more tricks are needed (Du et al., 2021).

**Data-initialized LD for density estimation.** Furthermore, in addition to the question of bias in Eq. 4 under MCMC, there is the question of variance. Hinton (1999) has argued that even after convergence, Monte Carlo estimates of the expected energy gradient will suffer from high variance because samples are being drawn from all over the model's (initially random) energy landscape. To remedy this, he proposed another initialization scheme (Hinton, 1999), namely, to start the Markov chains at *observed data*. This is certainly sensible late in training, when the model resembles the data, and such an initialization will shorten the burn-in period. But the procedure can also be justified as the approximate gradient of a different loss function, a difference or contrast of KL divergences, which has the same minimum as the original KL divergence (cf. Eq.

2) (Hinton, 2002). Accordingly, Hinton called data-initializing the Markov chain the method of "contrastive divergence" (CD). The *pairing* of each model sample with a data sample reduces variance (see Section A.3). However, contrastive divergence can also be used in conjunction with persistent chains; indeed, this is (to our knowledge) where persistent chains originated (Tieleman, 2008).

The approximation in CD comes from neglecting a term that is supposedly small in practice (Hinton, 2002), and the recent work of Du et al. (2021) has argued that this term can and should be modeled. But there are several reasons to believe this is not the case. First, the training scheme used by Du et al. (2021) is not in fact (persistent) contrastive divergence, but rather the *noise*-initialized persistent Markov chains lately described, so it is not clear what can be concluded about CD from these experiments.

Second, recent work has shown that, in certain settings, parameter estimation with contrastive divergence is consistent, asymptotically converges at close to the optimal rate, and approaches the optimal precision (the Cramér-Rao bound) (Jiang et al., 2018; Glaser et al., 2024). The results have so far been shown only for EBMs in which the parameters enter the energy linearly, but this nevertheless suggests there is nothing intrinsically problematic about CD as a density estimator.

Third and perhaps most compellingly, CD can be justified from other perspectives that do not invoke any approximation at all. For example, Omer & Michaeli (2021) show that CD can be seen as an instance of (conditional) noise-contrastive estimation (Gutmann & Hyvärinen, 2012). In this scheme, the Markov chain initialized at data is interpreted as a generator for a "negative-sample" distribution. With this generator fixed, the EBM is trained so as to make the initializing data (positive samples) and the samples from $L$ steps down the Markov chain (the negative samples) maximally distinguishable, which requires the EBM to become a better model for the data. Since the generator is in fact derived from the model, this in turn makes the discrimination problem more challenging, leading to more efficient training. Crucially, this interpretation retains the requirement that model and data samples be paired, which is critical for keeping the variance of the gradient small.

**Noise-initialized LD for generation.** Our hypothesis, then, is that CD is in fact a powerful density estimator, even for complex EBMs, but that this has been overlooked in the literature in part because CD-trained models do not generate high-quality images. That is because, conceptually, density estimation and (image) generation are to some extent orthogonal demands. Density estimation requires assigning the right probabilities to data, but what probabilities should be assigned to the noise vectors used to initialize sampling? Indeed, it is not clear that it makes sense to assign probabilities to data that are so far from the manifold to which the training data are confined; and if the data distribution has no support in this region, the gradient is zero. But if we want to generate samples, we must shape the energy landscape between the "noise manifold" and the data manifold.

On the other hand, non-convergent short-run MCMC models have shown how to shape the path from noise manifold to data manifold. This training procedure, although originally derived as an attempt at maximum-likelihood estimation, i.e. following the gradient in Eq. 2, is perhaps best understood as training a Wasserstein GAN in which the generator and discriminator are different aspects of the same model (Xiao et al., 2021b). In particular, the (negative) energy acts as a discriminator between data and samples produced by a *generator*, which in this case is LD operating on this same energy (Eq. 3), and initialized at noise. This encourages the EBM to assign lower energies to data samples than to noise (the discriminator); but also for the

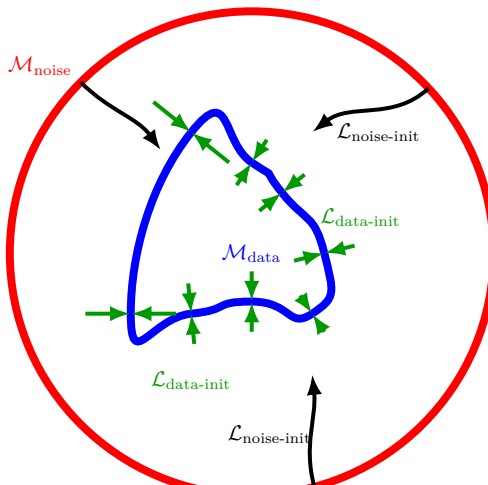

Figure 1: Illustration of the hypothesized roles of the two loss functions in shaping the energy landscape. The loss based on data-initialized Markov chains (i.e., CD) ensures that the data manifold has lower energy than surrounding points. The loss based on noise-initialized Markov chains ensures that the energy decrease from the noise manifold toward the data manifold.

---

**Algorithm 1:** Hybrid training of EBM

---

```
 1 def OneStepUpdate(
 2     x^(1),…,x^(N)        // batch of training data
 3     θ                   // current parameters of the EBM
 4     L,                  // number of Langevin steps
 5     ε,                  // size of Langevin step
 6     T,                  // temperature of Langevin dynamics
 7     η,                  // size of gradient-descent step
 8     λ,                  // relative weight of CD loss
 9 ):
```

$$10 \quad \boldsymbol{g}_{\text{data}} \leftarrow \texttt{AvgEnergyGradient}\big(\boldsymbol{x}^{(1)},\ldots,\boldsymbol{x}^{(N)},\boldsymbol{\theta}\big)$$

$$11 \quad \hat{\boldsymbol{x}}_L^{(0)},\ldots,\hat{\boldsymbol{x}}_L^{(N)} \leftarrow \texttt{LangDyn}\big(\boldsymbol{x}^{(1)},\ldots,\boldsymbol{x}^{(N)},\boldsymbol{\theta},L,\epsilon,T\big)$$

$$12 \quad \boldsymbol{g}_{\text{data-init}} \leftarrow \texttt{AvgEnergyGradient}\Big(\hat{\boldsymbol{x}}_L^{(1)},\ldots,\hat{\boldsymbol{x}}_L^{(N)},\boldsymbol{\theta}\Big)$$

$$13 \quad \hat{\boldsymbol{x}}_0^{(1)},\ldots,\hat{\boldsymbol{x}}_0^{(N)} \sim \mathcal{N}(\mathbf{0},\mathbf{I})$$

$$14 \quad \hat{\boldsymbol{x}}_L^{(0)},\ldots,\hat{\boldsymbol{x}}_L^{(N)} \leftarrow \texttt{LangDyn}\Big(\hat{\boldsymbol{x}}_0^{(1)},\ldots,\hat{\boldsymbol{x}}_0^{(N)},\boldsymbol{\theta},L,\epsilon,T\Big)$$

$$15 \quad \boldsymbol{g}_{\text{noise-init}} \leftarrow \texttt{AvgEnergyGradient}\Big(\hat{\boldsymbol{x}}_L^{(1)},\ldots,\hat{\boldsymbol{x}}_L^{(N)},\boldsymbol{\theta}\Big)$$

$$16 \quad \boldsymbol{\theta} \leftarrow \texttt{ParamUpdate}(\boldsymbol{g}_{\text{data}},\boldsymbol{g}_{\text{data-init}},\boldsymbol{g}_{\text{noise-init}},\boldsymbol{\theta},\eta)$$

```
17     return θ
18
```

$$19 \quad \texttt{def LangDyn}(\hat{\boldsymbol{x}}_0^{(1)},\ldots,\hat{\boldsymbol{x}}_0^{(N)},\boldsymbol{\theta},L,\epsilon,T):$$

```
       // in practice, vectorize
20     for n = 1,…,N do
```

$$21 \quad \hat{\boldsymbol{x}} \leftarrow \hat{\boldsymbol{x}}_0^{(n)}$$

```
22         for l = 1,…,L do
```

$$23 \quad \hat{\boldsymbol{z}} \sim \mathcal{N}(\mathbf{0},\mathbf{I})$$

$$24 \quad \hat{\boldsymbol{x}} \leftarrow \hat{\boldsymbol{x}} - \epsilon\frac{\partial E}{\partial \hat{\boldsymbol{x}}}(\hat{\boldsymbol{x}},\boldsymbol{\theta}) + \sqrt{2\epsilon T}\hat{\boldsymbol{z}}$$

$$25 \quad \hat{\boldsymbol{x}}^{(n)} \leftarrow \hat{\boldsymbol{x}}$$

$$26 \quad \texttt{return } \hat{\boldsymbol{x}}^{(1)},\ldots,\hat{\boldsymbol{x}}^{(N)}$$

```
27
```

$$28 \quad \texttt{def AvgEnergyGradient}(\boldsymbol{x}^{(1)},\ldots,\boldsymbol{x}^{(N)},\boldsymbol{\theta}):$$

$$29 \quad \texttt{return } \frac{1}{N}\sum_{n=1}^{N}\frac{\partial E}{\partial \boldsymbol{\theta}}(\boldsymbol{x}^n,\boldsymbol{\theta})$$

```
30
```

$$31 \quad \texttt{def ParamUpdate}(\boldsymbol{g}_{data},\boldsymbol{g}_{data\text{-}init},\boldsymbol{g}_{noise\text{-}init},\boldsymbol{\theta},\eta):$$

```
32     return
```

$$\boldsymbol{\theta} - \eta[(\boldsymbol{g}_{\text{data}} - \boldsymbol{g}_{\text{noise-init}}) + \lambda(\boldsymbol{g}_{\text{data}} - \boldsymbol{g}_{\text{data-init}})]$$

---

LD to transform noise samples into data samples (the generator). Together these demands oblige the energy to decrease from the "noise manifold" toward the data manifold.

It is not a coincidence that (self-)adversarial losses provide convenient interpretations for noise-initialized sampling as well as data-initialized sampling. This is what licenses writing the so-called (Nijkamp et al., 2020)"computational loss" for each as a difference in expected energies,

$$\mathcal{L}(\boldsymbol{\theta}) := \Big\langle E(\boldsymbol{X},\boldsymbol{\theta})\Big\rangle_{p(\boldsymbol{X})} - \Big\langle E(\hat{\boldsymbol{X}},\boldsymbol{\theta})\Big\rangle_{\Pi_{\text{gen}}(\hat{\boldsymbol{X}})}, \tag{5}$$

where $\Pi_{\text{gen}}(\hat{\boldsymbol{X}})$ is the distribution of data produced by the generator: Since for adversarial losses, the generator is fixed during improvement of the discriminator, derivatives with respect to the parameters $\boldsymbol{\theta}$ can pass inside the second sample average, even though the generator itself—temporarily fixed—also depends on these parameters.

**A hybrid loss function.** If density estimation and generation are indeed distinct demands, then to achieve both in one model will require separately enforcing both. Accordingly, we propose a hybrid loss combining the two schemes just described (see Fig. 1): a data-initialized version of Eq. 5, which enforces a self-adversarial noise-contrastive loss for modeling the data distribution; and a noise-initialized version of Eq. 5, which enforces a self-adversarial WGAN loss for generating from noise:

$$\mathcal{L}(\boldsymbol{\theta}) := \left[\Big\langle E(\boldsymbol{X},\boldsymbol{\theta})\Big\rangle_{p(\boldsymbol{X})} - \Big\langle E(\hat{\boldsymbol{X}},\boldsymbol{\theta})\Big\rangle_{\Pi_{\text{noise-init}}(\hat{\boldsymbol{X}})}\right] + \lambda\left[\Big\langle E(\boldsymbol{X},\boldsymbol{\theta})\Big\rangle_{p(\boldsymbol{X})} - \Big\langle E(\hat{\boldsymbol{X}},\boldsymbol{\theta})\Big\rangle_{\Pi_{\text{data-init}}(\hat{\boldsymbol{X}})}\right]. \tag{6}$$

The relative weighting of the losses, $\lambda$, is a hyperparameter and must be tuned. We emphasize the simplicity of the loss gradient: although the more exact interpretation is in terms of self-adversarial losses, it amounts to combining the pseudo-MLE gradient (i.e., short-run MCMC) with the contrastive-divergence gradient. The training algorithm is summarized in Algorithm 1.

## 3  Methods

**Synthetic dataset.** We illustrate the main ideas on a synthetic dataset designed to replicate settings where the data lie on a low-dimensional manifold embedded in a higher-dimensional ambient space. Specifically,

the dataset consists of three concentric rings of radius 1, 2 and 3 with equal probability in a 2D ("X-Y") plane. To each point on the ring, we add isotropic Gaussian noise with standard deviation 0.1 in the X-Y-Z plane (i.e., in the original X-Y plane as well as one orthogonal dimension); and 0.01 along each of seven additional orthogonal dimensions, resulting in a dataset with an intrinsic 2D structure embedded in a 10D ambient space (see Section A.1 for full details).

We investigate five different training schemes, each based on different initializations of the Langevin dynamics, including the hybrid approach just proposed; these are detailed in the next section. For all experiments, we use a four-layer ConvNet architecture. For persistent and persistent+refresh initializations, we apply Langevin dynamics with a step size of $\epsilon = 0.05$ and a temperature (see Section A.4) of $T = 0.005$. For data and hybrid initializations, we employ an adaptive step size: at each training iteration, the step size is set as $\epsilon = 0.0005/||E(x)||$, allowing for larger exploration steps early in training and gradual annealing as the training progresses. All models were trained for 10,000 iterations using a single V100 GPU. We experimented with using adaptive step size for the persistent and noise-init; however, we found best results with constant step size. To generate KDE plots, we initialize the MCMC chains from standard normal noise, using the same Langevin parameters employed during training.

**Natural datasets.** We also apply these five training schemes to CIFAR-10 and the Oxford Flowers dataset. We employ an architecture similar to the one detailed by Nijkamp et al. (2019), with their $n_f$ parameter set to 128; this translates to 1024 output channels in the final convolutional layer. We found that incorporating attention layers significantly enhances the learning of convergent models in these more complex datasets, compared to merely stacking convolutional layers. A similar observation was reported by Nijkamp et al. (2019). We train these models with the same five schemes as for the synthetic dataset, in this case for 50,000 iterations and again each on a single V100. We apply the same MCMC parameters across all schemes, setting the step size $\epsilon = 1$ and temperature $T = 5e-5$ (as for the synthetic dataset), and taking $L = 500$ steps of Langevin dynamics.

At test time, "short-run" samples are generated by initializing Langevin dynamics at standard-normal noise samples and taking the same number of steps as during training, $L = 500$. The exception is the hybrid model, for which $L = 1000$ and $\epsilon = 2$ yields slightly better looking samples (it does not improve any other models). We also continue these chains for 100,000 steps to investigate their long-run behavior. *Data-initialized* chains are run for $L = 10,000$ steps, starting from samples from the *test* partition of the relevant data sets. All models are initialized at the same test-data samples to facilitate comparisons between them.

**Common training and testing details.** For the hybrid loss we set $\lambda = 25$ for both datasets (see Eq. 6). We tried a range of values for $\lambda$ and found 25 to give the best image generation.

In persistent training, we initialize a sample bank of size 60,000 with random noise. During each training iteration, we randomly select a batch from this sample bank, run a Markov chain starting from that batch, and replace the selected batch with the resulting samples. Following Du et al. (2021), in the "refresh" variant, we use this procedure only 95% of the time. In the remaining 5% of iterations, instead of starting from a batch in the sample bank, we "refresh" the bank by initializing MC chains from pure noise and then replacing a randomly selected batch in the bank with the newly generated samples.

**Out-of-distribution detection.** We follow the standard procedure (Hendrycks & Gimpel, 2017) to determine how well the model separates the probability densities of two datasets, in this case CIFAR-10 (on which EBM was trained) and SVHN. In particular, we first calculate the energies for all images in the test sets of CIFAR-10 and SVHN. We then calculate the fractions of true and false positives for every choice of an energy threshold putatively separating the two datasets. Finally, we report the area under the graph of true positives vs. true negatives, i.e. the area under the receiver-operator characteristic (AUROC).

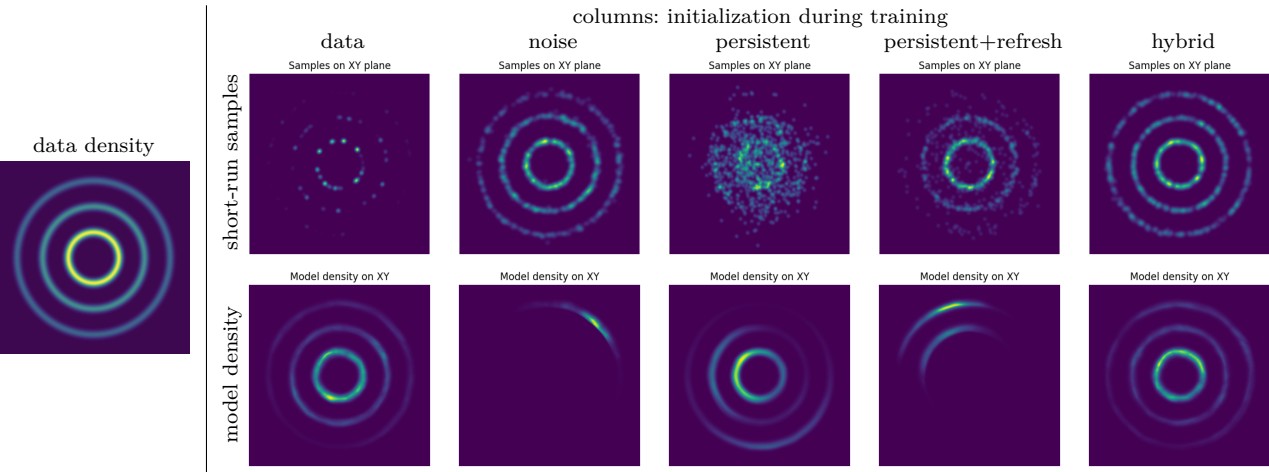

Figure 2: Data distribution (left) and learned densities (right) under different EBM training schemes. Top row shows a kernel-density estimate of the distribution learned by the sampler (Eq. 3); bottom row shows true model densities (Eq. 1).

## 4 Results

### 4.1 Synthetic (rings) dataset

Natural data, like images, are believed to be confined to a manifold of much lower dimension than the ambient (pixel) space (Bengio et al., 2013). To explore such a setting without losing the ability to visualize that manifold, we begin with a 10-dimensional distribution with most of its mass concentrated in concentric rings in a two-dimensional subspace (see the leftmost panel in Fig. 2, the description in Section 3, and a more detailed description in Section A.1). The extra eight dimensions make the task of finding the data manifold from noise more challenging. It is not possible to visualize the entire energy landscape learned by the model, but we can view the space of most interest, the "X-Y" plane in which the probability mass is concentrated. We also examine kernel density estimates in this 2D space of samples generated by short-run MCMC. As Nijkamp et al. (2019; 2020) have pointed out, these distributions need not match.

We trained identical EBMs (see Section 3) using five distinct schemes (columns at right in Fig. 2): (1) initialization from data (i.e., contrastive divergence); (2) initialization from noise; (3) persistent initialization (à la Nijkamp and colleagues, i.e. with chains originally initialized at noise rather than data); (4) persistent initialization with 5% of chains randomly restarted at noise; and (5) our hybrid loss, Eq. 6.

Training models with data-initialized Markov chains (Fig. 2, first column) accurately shapes the energy landscape on and very near the data manifold, as seen in the learned model density (bottom row). On the other hand, the Markov chains run during training tend to remain close to the data manifold and fail to explore distant regions of the space. Since these regions of the energy landscape are not visited during training, they are not (appropriately) modified by gradient descent either. Consequently, at test time, Markov chains initialized at standard noise struggle to reach the data manifold (top row).

In contrast, models trained with noise-initialized Markov chains (second column) learn to generate from the data distribution (top row), but do not actually learn the correct energy (bottom row), as noted by Nijkamp and colleagues. Allowing chains to persist across weight updates (third column) improves density estimation near the data (bottom row), but at the price of poorer generation from noise (top row). The latter problem, in turn, can be partially ameliorated by randomly restarting 5% of the persistent chains from noise at every step of training—but at the price of unbalancing coverage of the data (bottom row).

Finally, EBMs trained with our hybrid loss (Eq. 6) combine the merits of the models trained with noise- and data-initialized MCMC: generating new samples from noise in virtue of the former, and assigning correct probabilities on the data manifold in virtue of the latter. The important empirical finding is that, although

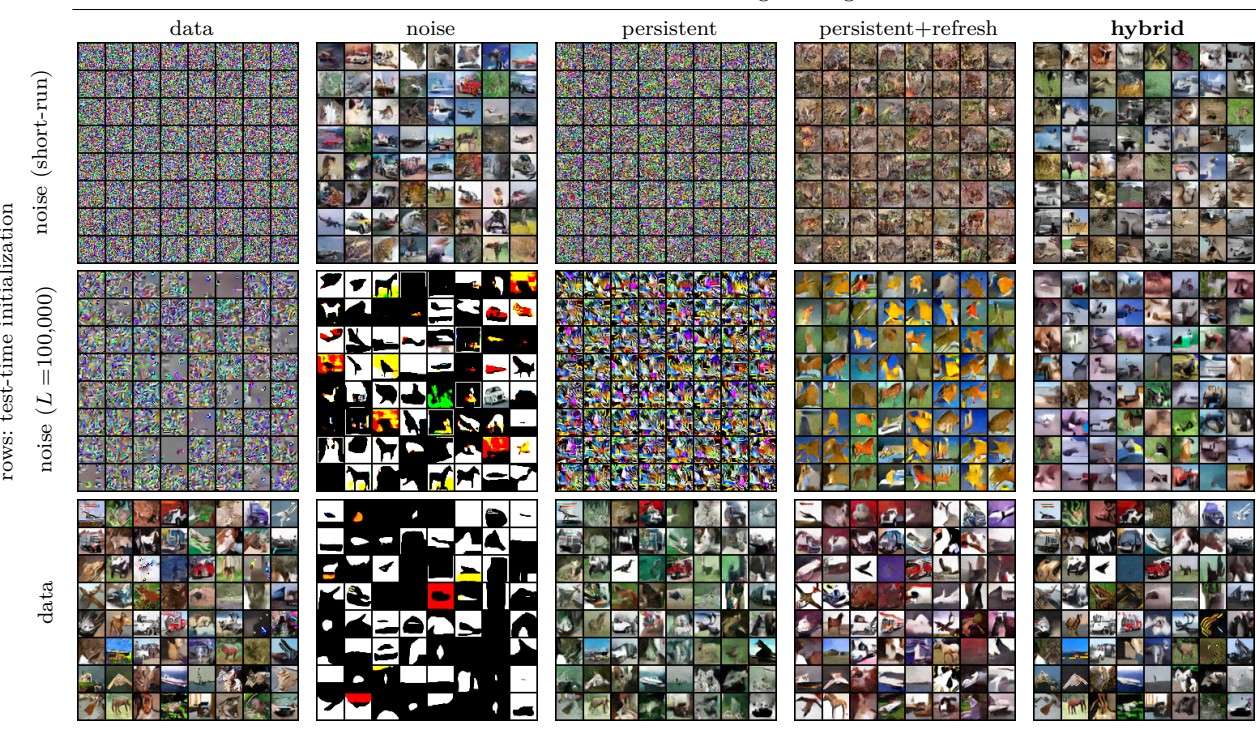

Figure 3: Data generation for networks trained on CIFAR-10. Each column shows the results from a model trained with a particular initialization scheme. Each row shows samples generated at test time with Langevin dynamics (LD). First row: LD initialized at noise and run for hundreds of steps ("short-run"). Second row: LD initialized at noise and run for 100,000 steps ("long run"). Third row: LD initialized at data and run for 10,000 of step. For ease of comparison, all models were initialized at the *same* data samples.

the two component losses impose at least partially distinct constraints, they do not interfere with each other; and together they cover the relevant portions of the energy landscape.

## 4.2 Natural datasets

We now ask if these findings extend to more complicated data distributions, to wit, CIFAR-10 and the Oxford Flowers datasets. As with our experiments on the rings dataset, we trained otherwise identical models using the same five distinct approaches. It is obviously not possible to view the probability densities of these distributions, so we instead examine individual samples from the distributions of interest (Figs. 3 and 4) and the energies assigned by models to selected samples (Fig. 5); and then quantify performance in terms of FID score (Table 1) and out-of-distribution detection (Table 2).

### 4.2.1 Qualitative results

We begin by reproducing the results of Nijkamp et al. (2019; 2020), namely, that EBMs trained with noise-initialized Markov chains do not produce convergent Langevin dynamics at test time. In such models, energy continues to decline as long as the Markov chain is run, at least out to $L = 10,000$ steps (Fig. 9). Under the other four training schemes, Markov chains do converge (Fig. 9). But do they converge to the data distribution?

We generate samples from the models in three different ways: short-run MCMC from noise, to verify whether the model can be used to generate images; long-run MCMC from noise, to determine to what (if anything) this sampler converges; and MCMC from data, to determine the quality of the models near the data.

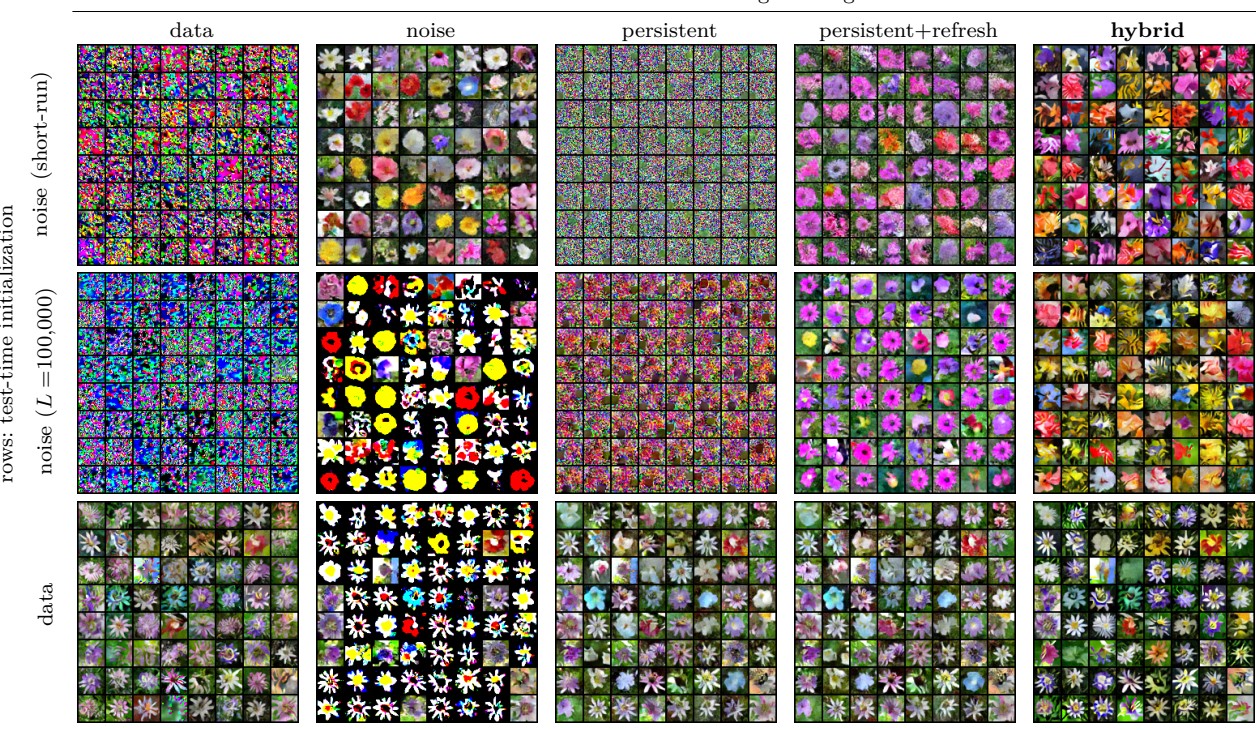

Figure 4: Data generation for networks trained on the Oxford Flowers. Each column shows the results from a model trained with a particular initialization scheme. Each row shows samples generated at test time with Langevin dynamics (LD). First row: LD initialized at noise and run for hundreds of steps ("short-run"). Second row: LD initialized at noise and run for 100,000 steps ("long run"). Third row: LD initialized at data and run for 10,000 of step. For ease of comparison, all models were initialized at the *same* data samples.

As with the rings data set, models trained on CIFAR-10 and Oxford Flowers purely with contrastive divergence (i.e., data-initialized Markov chains, first column of Figs. 3 and 4) learn excellent models of the data: Chains initialized at test (unseen) images yield high-quality samples even after $L = 10,000$ LD steps. But the model cannot generate images from noise, because during training it never explored regions far from the data manifold. This is true in the long run (middle row) as well as the short run (top row); that is, Langevin dynamics does not ever converge on the data distribution. Indeed, this is likely the reason that contrastive divergence as introduced by Hinton has not been used to train EBMs—despite apparently learning very good models of the data. Even when the model has learned to assign correct energies on and near the data manifold, it need not have shaped the energy landscape in vast regions of the ambient space. For example, there is no reason why a vector of standard normal noise should have higher or lower energy than a nearby vector that is closer to the data manifold—so the energy gradient in Eq. 3 is not helpful. And since the true data manifold has a much lower intrinsic dimension than the ambient space, most steps of LD are orthogonal to the manifold (so the noise term in Eq. 3 does not help either).

As has been observed in the literature, noise initialization (second column in Figs. 3 and 4) yields a generator of high-quality images (top row), but the model has not learned the correct densities, since the sampler moves toward saturated images—both under data-initialization (bottom row) and in the long run under noise initialization (middle row). Again we see that persistent initialization (third column) improves the model density near data (bottom row), but at the price of being unable to generate new samples from noise (top and middle rows). This also shows that the energy convergence observed in Fig. 9, even coupled with a good model near the data, cannot by itself be taken to imply convergence to the data distribution. Indeed, the energies of the (poor) images generated by long-run, noise-initialized chains are similar to the energies of images generated by the data-initialized chains (Fig. 5, "persistent"), so the resulting model is evidently

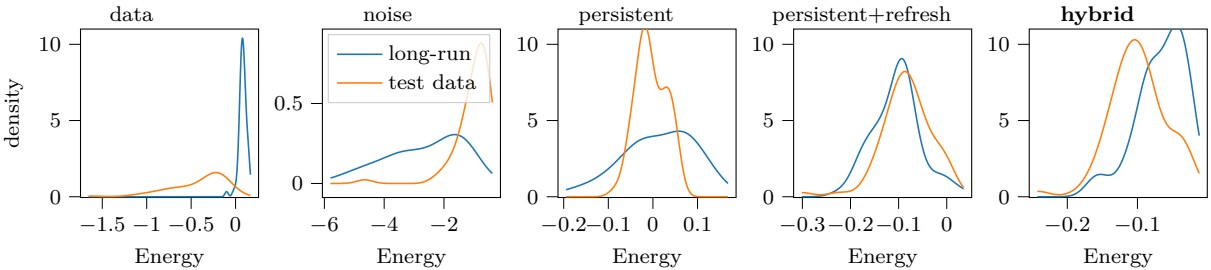

Figure 5: Distributions of energies assigned by models after training on CIFAR-10. Blue: long-run samples; orange: test (held-out) CIFAR-10 samples. Plot titles specify the MCMC initialization scheme used during training.

not a good density estimator. (Fig. 5 shows the distribution of energies for CIFAR-10; for Oxford flowers, see Fig. 10 in the appendix.)

During training, occasionally initializing the Markov chains from noise rather than from the sample bank (fourth column of Figs. 3 and 4; see Section 3), as proposed by Du & Mordatch (2019), does markedly improve sampling from noise (first and second rows). However, the resulting samples are either unacceptably poor (Fig. 3) or concentrated in one or two modes of the distribution (Fig. 4). It is true that this method was not intended to be used with noise-initialization; we have already argued that this is a limitation of such models. But furthermore, like its counterpart without sample-bank refreshing, this model assigns very similar energies to the (poor) samples generated from long-run, noise-initialized chains as to (good) samples from data-initialized chains (Fig. 5). So the model is still not a good density estimator.

As with the synthetic dataset, if we do desire to generate samples from noise, we need to shape the energy in this region of pixel space as well—in the present case, essentially an $n$-sphere with radius $\sqrt{N_{\mathrm{dims}}}$, since we use standard normal noise and the space is high-dimensional. In particular, the energy must decrease roughly monotonically from this noise manifold toward the data manifold. Combining the data-initialized and noise-initialized losses during training (Eq. 6) again yields a model (final column of Figs. 3 and 4) that can generate high-quality samples from noise in the short run (top row) and converges to such samples in the long run (middle row), without compromising the energy landscape near the data (bottom row). The energies assigned to long-run samples are very close to, but slightly higher than, the energies assigned to test samples (Fig. 5, final panel). Although we cannot rule out the existence of other points with even lower energy, this strongly suggests that models trained with the hybrid loss assign the highest probability to images resembling the training dataset. Thus, models trained with the hybrid loss make good generators as well as density estimators.

### 4.2.2 Quantitative results

To quantify the image generation from the previous section we compute FID scores (Heusel et al., 2017) for CIFAR-10 for the five training schemes (Table 1), generating samples from each with short-run LD initialized at noise. Our hybrid-trained EBMs ($\lambda = 25$) achieve the lowest scores, lower even than models trained with noise initialization. Thus, so far from compromising image generation from noise, the addition of the data-initialized loss term improves it. Of course, performance is not monotonically increasing in $\lambda$, since in the limit, the hybrid scheme becomes contrastive-divergence training (i.e., pure data initialization, the first row of Table 1). By about $\lambda = 50$, hybrid training becomes worse than pure noise initialization, presumably because the path from noise to data is no longer well-learned.

Table 1: FID scores for CIFAR-10.

| test-time init | $L_{\text{test}}$ | training scheme | FID |
|---|---|---|---|
| noise | short | data | 292 |
| | | noise | 42.3 |
| | | persistent | 243 |
| | | persistent+refresh | 94 |
| | | hybrid, $\lambda = 10$ | 38.7 |
| | | **hybrid, $\lambda = 25$** | **36.8** |
| | | hybrid, $\lambda = 50$ | 44.5 |
| noise | long | hybrid, $\lambda = 25$ | 35.3 |
| sample bank | short | persistent | 40.1 |
| | | **persistent+refresh** | **35.1** |

Note that if we continue to run the Langevin dynamics in the hybrid-trained EBMs for a very long time ($L_{\text{test}} = 100,000$), the FID score does not much change (middle row of Table 1, FID = 35.3). This is consistent with the observation that the short- and long-run samples from this model look qualitatively similar to each other (Fig. 3, last column, top and middle rows), and corroborates the conclusion that short-run LD convergences to the stationary distribution in EBMs trained with the hybrid loss, whereas it does not under the noise-initialized loss and its variants.

For comparison, we also include the performance of EBMs trained with persistent Markov chains and then tested by initializing LD at the accumulated bank of samples rather than noise (bottom row of Table 1). Even when initialized at a bank of high-quality samples, these models achieve at best comparable FID scores to the hybrid model initialized at noise.

The complementary question, how well the model has learned the underlying density, can be quantified indirectly by asking how well the distribution of model energies for samples from CIFAR-10 can be distinguished from the same model's energies for samples from another dataset, in this case SVHN (Netzer et al., 2011). Following the literature (Hendrycks & Gimpel, 2017), we report the area under the receiver-operator characteristic (AUROC), for the training schemes we have been considering so far, in Table 2. We include models trained with our hybrid loss for various values of the relative weighting $\lambda$, including $\lambda = 25$, used throughout up until this point. Even at $\lambda = 10$, models trained with the hybrid loss outperform models trained with noise- and persistently initialized Markov chains. In fact, performance increases as a function of $\lambda$, achieving maximal values for pure CD (all weight on the second component of the loss in Eq. 6). This suggests that including noise-initialized Markov chains in the loss function does degrade density estimation by some small amount—the price to be paid for generating new samples from noise.

Table 2: Out-of-distribution performance for models trained on CIFAR-10 and tested on CIFAR-10 and SVHN. AUROC = Area under ROC curve.

| MCMC init | AUROC |
|---|---|
| **data** | **0.95** |
| noise | 0.57 |
| persistent | 0.68 |
| persistent+refresh | 0.66 |
| hybrid, $\lambda = 10$ | 0.75 |
| hybrid, $\lambda = 25$ | 0.87 |
| hybrid, $\lambda = 50$ | 0.91 |

That OOD detection improves monotonically with the weight on the data-initialized loss ($\lambda$), but FID does not, is intuitive and related to the fact that in general we require $\lambda \gg 1$, i.e. that the data-initialized loss be weighted much more highly than the noise-initialized loss. The EBM needs to learn a more detailed model of the data than of the noise distribution, or of the path from noise to data, since it does not matter much how the sampler arrives at the data manifold, as long as it does; whereas generating high-quality images or assigning the right probabilities to data requires many more bits. However, this is not the only reason that $\lambda \gg 1$ is required for good performance: it also decreases the variance of the gradient, which we show next.

### 4.2.3 Evolution of the computational losses

Recall that the "computational losses" defined in Eqs. 5 and 6 treat the generator $\Pi(\hat{\boldsymbol{X}})$ as fixed (i.e., a "stop-gradient" operation is applied to them). As a result, these equations do not correspond to genuine loss functions. In particular, we do not expect them to decline roughly monotonically throughout training, as genuine loss functions do, because stepping downhill under the pretense of a fixed generator will in fact change the generator and may cause the total loss to increase. Instead, the functions in Eqs. 5 and 6 correspond to loss *gradients* (with respect to a fictive coldness parameter; see Section A.3), and so we expect them to decline toward zero and then oscillate. Nevertheless, large fluctuations in these updates indicate instability and are undesirable.

We invoked at the outset (Section 2) Hinton's 1999 argument that data-initializing the sampler reduces variance in the estimate of the gradient. This explains why our method is successful while the superficially similar training based on persistent sample banks with random restarts at noise (Du & Mordatch, 2019; Du et al., 2021) is not: The opposing average energy gradients in Eq. 2 can be "paired" in CD—both are "initialized" at the same data, with the second gradient evaluated farther down the Markov chains. In contrast, samples from a persistent bank will eventually be near the data manifold, but not necessarily near precisely those data showing up in the first (data-averaged) energy gradient. That is, the variance of the

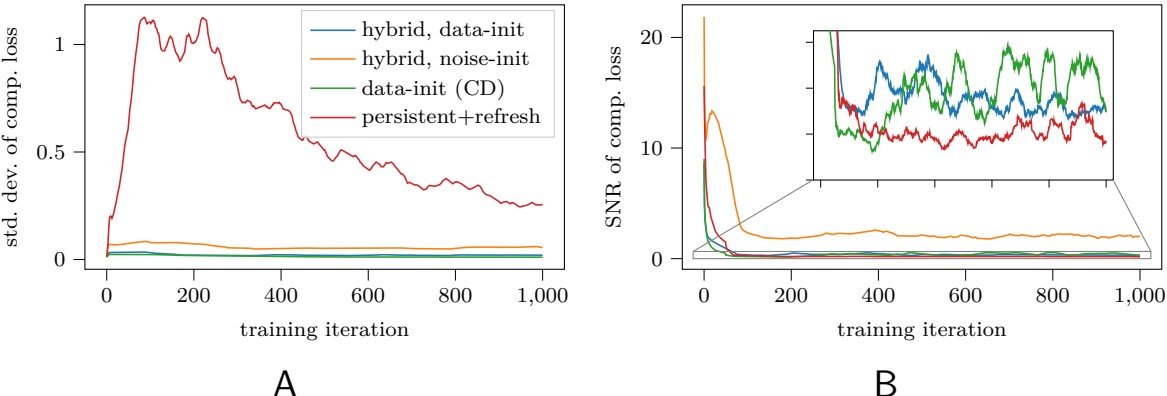

Figure 6: The evolution of the computational losses and their components (see legend) across training on CIFAR. (A) Standard deviations. (B) SNR (mean to standard deviation). Curves have been smoothed for clarity.

computational loss is

$$\mathrm{Var}[\mathcal{L}] = \mathrm{Var}\left[\frac{1}{N}\sum_{n=1}^{N}E(\boldsymbol{X}^{(n)},\boldsymbol{\theta}) - \frac{1}{N}\sum_{n=1}^{N}E(\hat{\boldsymbol{X}}^{(n)},\boldsymbol{\theta})\right]$$
$$= \frac{1}{N^2}\sum_{n=1}^{N}\left(\mathrm{Var}\left[E(\boldsymbol{X}^{(n)},\boldsymbol{\theta})\right] + \mathrm{Var}\left[E(\hat{\boldsymbol{X}}^{(n)},\boldsymbol{\theta})\right] - 2\mathrm{Cov}\left[E(\boldsymbol{X}^{(n)},\boldsymbol{\theta}),E(\hat{\boldsymbol{X}}^{(n)},\boldsymbol{\theta})\right]\right),$$

with $N$ the batch size. When $\boldsymbol{X}^{(n)}$ and $\hat{\boldsymbol{X}}^{(n)}$ are independent (for all $n$), as in persistent or noise initializations, the covariance term vanishes. Under data initialization, each sample $\hat{\boldsymbol{X}}^{(n)}$ is generated from a Markov chain that starts at $\boldsymbol{X}^{(n)}$. If the Markov chain is not run past the burn-in stage, then the samples and their corresponding energies (under the model) will be positively correlated, thus reducing the overall variance.

We can observe this empirically by plotting the standard deviation (across a batch) of the computational loss at each training iteration (Fig. 6A). As expected, the estimates of the computational loss have the highest variance under persistent initialization, and the lowest under data initialization (CD). Under our hybrid scheme, both components' deviations are much closer to that of CD. The noise-initialized loss is larger, as expected, but recall that during training, it is effectively scaled down by a factor of $\lambda = 25$ relative to the data-initialized loss.

It might be argued that the relative, rather than absolute, variability of the computational loss is the critical quantity. But the same conclusion holds: persistent initialization also yields the loss estimates with the lowest signal-to-noise (mean-to-standard-deviation) ratio (Fig. 6B, especially the inset). (The highest SNR estimates of the loss are found in the noise-initialized component of the hybrid loss, but this is arguably misleading since the energy need not be estimated to high precision far from the data manifold.)

### 4.2.4 Sample diversity

Another failure mode of generative models is the inability to generate *novel* samples. For example, EBMs trained with persistent Markov chains generate higher-quality examples at test-time when initialized at samples from a bank (accumulated during training) rather than at noise samples (recall Table 1). But the price to be paid is that, at least without more "tricks," a generated sample will frequently be very similar to the sample from the bank at which the LD was initialized.

We show this in Fig. 7. Fig. 7A shows a sample bank accumulated during training of an EBM on the Oxford Flowers, and Fig. 7B shows the samples generated from the trained EBM after $L = 500$ steps of LD, starting form the sample bank. The resemblance between Fig. 7A,B is very strong; we might even say that the "generation" process is mostly just copying samples out of the bank.

In contrast, EBMs trained under the hybrid loss generate novel samples. Fig. 7C shows samples from such an EBM in the first column. The next three columns show their (pixelwise) nearest neighbors in the training set. Evidently, the generated images do not closely resemble the training data, which shows that the EBM has learned the structure of (CIFAR-10) images rather than merely memorizing examples.

### 4.2.5 Metropolis adjustment and the stationary distribution

Theoretically, to ensure that Langevin sampling has the correct stationary distribution despite errors introduced by the discretization (see Section 1), some samples must be rejected (Metropolis et al., 1953). In particular, samples can be "accepted" according to a formula

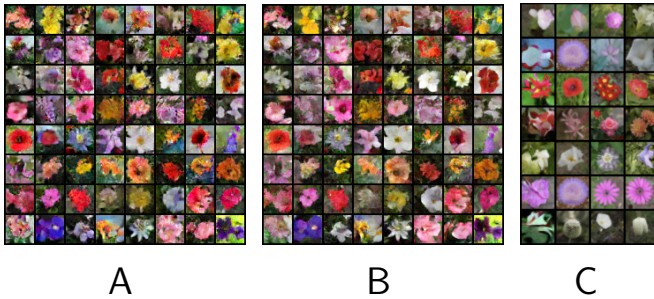

Figure 7: Sample diversity. (A) A bank of samples accumulated during training of an EBM with persistent Markov chains (with periodic refreshing). (B) Samples generated from this EBM after training, using short-run Langevin dynamics initialized at the samples in (A). (C) The first column shows images generated by an EBM trained under the hybrid loss; the remaining columns show their nearest neighbors in the training set.

that guarantees that the Markov chain satisfies detailed balance, and consequently has the correct stationary distribution. However, when using LD to sample from EBMs, the acceptance rate typically drops to very small values, and the sampling procedure slows to a crawl (rejected samples do not count toward Langevin steps). So this "Metropolis adjustment" is frequently omitted, as we have until this point.

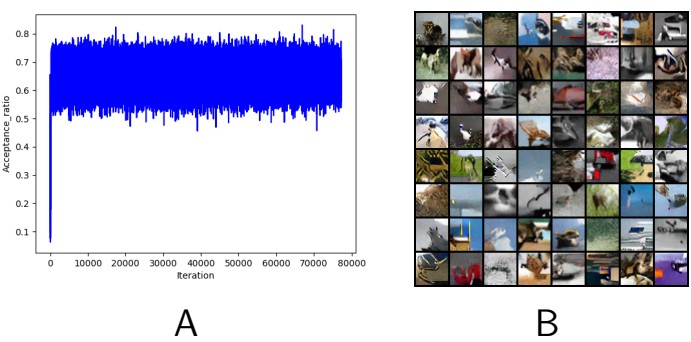

Figure 8: Image generation with the Metropolis adjustment. (A) The acceptance ratio as a function of training iterations. (B) Samples after 50,000 Langevin steps, Metropolis adjusted, from an EBM trained with the hybrid loss.

Nevertheless, to demonstrate more rigorously that the learned models do have the correct stationary distribution, we now apply the Metropolis adjustment. To keep the chain from "seizing," we replace the fixed step size $\epsilon$ with a dynamic step that decreases in proportion as the acceptance ratio falls below the optimal ratio of 0.67 (Neal, 2011). Here we use this procedure to generate samples, starting from noise, from an EBM trained with our hybrid loss on CIFAR. Fig. 8A shows that this technique does indeed maintain an acceptance ratio close to the optimum. Notice that we continue to sample two orders of magnitude beyond the number of LD steps taken during training. Fig. 8B shows samples from such chains after 50,000 LD steps. Critically, the resulting images are once again very high quality, and resemble the corresponding images generated without the Metropolis adjustment (Fig. 3, final column), both short-run (top row) and long-run (middle row). This strongly suggests that the learned model has the correct stationary distribution—i.e., that it is a good density estimator, as well as being a good generator.

## 5 Discussion

The main goal of this paper has been to revisit contrastive divergence as a training method and to understand why, despite its early successes with models such as restricted Boltzmann machines, it has become less popular for training modern energy-based models using neural networks. A crucial reason for this decline is

the current standard for evaluating EBMs, which primarily emphasizes the quality of images generated from known noise distributions. Since CD initializes training near the data manifold, it is inherently disadvantaged in generating paths from noise distributions, making it unreasonable to expect CD-trained models to excel at noise-to-data image generation.

In contrast, training schemes based on noise-initialized short-run MCMC, although able to produce visually appealing samples, have known shortcomings. Specifically, the images generated through such approaches do not represent samples from the stationary distribution of the trained model. Nijkamp et al. (2020) have previously discussed this issue and proposed remedies involving maintenance of a persistent sample bank (originally initialized from noise). However, our experimental findings on synthetic datasets—consisting of low-dimensional data embedded in high-dimensional spaces—demonstrate that persistent noise-initialized training often fails to recover the true underlying data distribution completely, whereas CD reliably succeeds. Furthermore, although persistent methods frequently showcase samples from stored filter banks, these approaches face significant difficulties in directly generating new images starting purely from noise distributions, as we have shown.

To address this challenge comprehensively, we introduced a simple yet highly effective hybrid training approach that combines the advantages of CD and noise-based methods. Specifically, we augment the traditional CD loss with a scaled MLE loss. This combined strategy ensures robust density estimation while simultaneously enabling high-quality image generation from noise. The procedure is simple and intuitive and does not require persistent chains, data augmentation, resetting Markov chains, or other expedients. Besides the two Langevin hyperparameters (number and size of steps), only the relative importance of the two penalties must be set beforehand, and we found a single value ($\lambda = 25$) to work well for all experiments. Increasing the weight on the data-initialized chain beyond this point (i.e., making training more CD-like) improves out-of-distribution detection (Table 2), but makes image generation worse (Table 1).

To distinguish cleanly the effects of the various training schemes, including ours, we have restricted their applications to EBMs based on a simple neural network and trained on correspondingly simple data sets. We expect scaling it to more complicated or higher-resolution images will require more work. Nevertheless, we believe that this hybrid training scheme offers a practical pathway to improve both density estimation and image-generation capabilities in EBMs.

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

# A   Appendix

## A.1   The concentric-rings datasets

The generative process for the concentric rings is depicted in the graphical model at right (we use $\tau := 2\pi$). The covariance matrix $\mathbf{D}$ is diagonal with $\sigma_{\text{big}}^2 := 0.1^2$ for the first three diagonal entries and $\sigma_{\text{small}}^2 := 0.01^2$ for the remaining seven. The three radius lengths are $r \in \{1.0, 2.0, 3.0\}$. We can visualize the distribution of $(X_1, X_2)$, that is, the "thick" parts of this distribution, by explicitly marginalizing out $Z$ and $\Phi$ (and ignoring the other dimensions of $\boldsymbol{X}$). Anticipating that the distribution should be invariant to the angle of $(X_1, X_2)$, we define the length $\ell := \sqrt{x_1^2 + x_2^2}$. Then we obtain

$$p(z) = 1/3 \qquad p(\phi) = \mathcal{U}(0, \tau)$$

$$p(\boldsymbol{x}|z, \phi) = \mathcal{N}\left( r_z \begin{bmatrix} \cos(\phi) \\ \sin(\phi) \\ \mathbf{0} \end{bmatrix}, \mathbf{D} \right)$$

$$p(x_1, x_2) = \sum_z \int_\phi p(x_1, x_2|\phi, z)p(\phi)p(z)\,\mathrm{d}\phi$$

$$\propto \sum_z \int_\phi \mathcal{N}\left( r_z \begin{bmatrix} \cos(\phi) \\ \sin(\phi) \end{bmatrix}, \sigma_{\text{big}}^2 \mathbf{I} \right)\mathrm{d}\phi$$

$$\propto \sum_z \int_\phi \exp\left\{ -\frac{\ell^2 - 2r_z(x_1 \cos\phi + x_2 \sin\phi) + r_z^2}{2\sigma_{\text{big}}^2} \right\}\mathrm{d}\phi$$

$$\propto \sum_z I_0\left( \frac{r_z\ell}{\sigma_{\text{big}}^2} \right) \exp\left\{ -\frac{\ell^2 - r_z^2}{2\sigma_{\text{big}}^2} \right\}$$

$$\propto \sum_z \tilde{I}_0\left( \frac{r_z\ell}{\sigma_{\text{big}}^2} \right) \exp\left\{ -\frac{\ell^2 - 2r_z\ell + r_z^2}{2\sigma_{\text{big}}^2} \right\}$$

$$\propto \sum_z \tilde{I}_0\left( \frac{r_z\ell}{\sigma_{\text{big}}^2} \right)\mathcal{N}\left( \ell; r_z,\ \sigma_{\text{big}}^2 \right),$$

with $\tilde{I}_0\left( \frac{r\ell}{\sigma_{\text{big}}^2} \right)$ a scaled version of the modified Bessel function of the first kind (to wit, `i0e` in `scipy`). Since $\sigma_{\text{big}} = 0.1$ is small compared to the inter-ring differences (1.0), $r_z\ell \approx r_z^2$. Thus, the distribution is approximately a one-dimensional Gaussian mixture model along any radius, with the Bessel functions providing the mixing weights.

## A.2   Energy evolution during Langevin dynamics

Fig. 9 shows the evolution of the energy across LD in models fully trained on CIFAR-10.

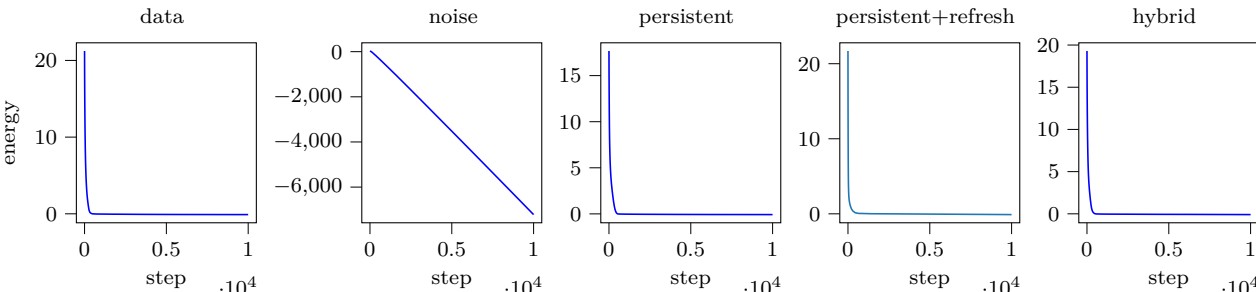

Figure 9: Energy vs. Langevin steps in models trained with various schemes, identified in the column labels, on CIFAR-10.

### A.3 The computational loss

Nijkamp et al. (2020) identify the difference in average energies,

$$d := \left\langle E(\boldsymbol{X}, \boldsymbol{\theta}) \right\rangle_{\boldsymbol{X}} - \left\langle E(\hat{\boldsymbol{X}}, \boldsymbol{\theta}) \right\rangle_{\hat{\boldsymbol{X}}}. \tag{7}$$

as a critical quantity during EBM training. (We leave open for now under precisely what distribution the second expectation is taken, but assume that it depends on the parameters $\boldsymbol{\theta}$.) They make, in particular, the empirical observation that $d$ and the fluctuations in the average magnitude of the force $\partial E/\partial \boldsymbol{x}$ correlate highly and positively at the same training batch, but negatively at lags of two or three batches. Thus stable training requires $d$ to balance roughly between positive and negative values across gradient updates.

**The computational loss is a gradient.** It is tempting to interpret Eq. 7 as the loss function itself because its gradient superficially resembles Eq. 2. But the averaging brackets in the second term of Eq. 7 depend on the parameters, so the gradient cannot pass inside. Nijkamp and colleagues point out this similarity to the loss but rightly resist the temptation to assimilate the two. But what then is the relationship?

Let us re-express the energy function (without loss of generality) as the product of a fixed-scale energy $\tilde{E} \in [-1, 1]$, and a "coldness" parameter $\beta > 0$: $E = \beta\tilde{E}$. This parameter could be explicit, or it could be merely a fictitious stand-in for the net effect of all the parameters on the energy magnitude. In either case, the loss gradient, Eq. 2, with respect to this parameter is

$$\frac{\mathrm{d}}{\mathrm{d}\beta} \mathrm{D}_{\mathrm{KL}}\{p(\boldsymbol{X}) \| \hat{p}(\boldsymbol{X}; \boldsymbol{\theta})\} = \mathbb{E}_{\boldsymbol{X}}\left[\frac{\partial E}{\partial \beta}(\boldsymbol{X}, \boldsymbol{\theta})\right] - \mathbb{E}_{\hat{\boldsymbol{X}}}\left[\frac{\partial E}{\partial \beta}(\hat{\boldsymbol{X}}, \boldsymbol{\theta})\right]$$

$$= \frac{1}{\beta}\left(\mathbb{E}_{\boldsymbol{X}}[E(\boldsymbol{X}, \boldsymbol{\theta})] - \mathbb{E}_{\hat{\boldsymbol{X}}}\left[E(\hat{\boldsymbol{X}}, \boldsymbol{\theta})\right]\right) = \frac{d}{\beta}.$$

Thus, the difference in expected energies is proportional to the gradient of the loss with respect to the energy magnitude.

This explains the relationships observed in (Nijkamp et al., 2020). If Eq. 7 were a proxy for the loss, gradient descent (with sufficiently small gradient steps) would always drive it downwards. Instead, $d$ is a loss *gradient*, so during successful training runs it oscillates around zero. Negative values of $d$ imply that the coldness $\beta$ needs to be adjusted upward by gradient descent. Equivalently, negative $d$ implies that the energy $E = \beta\tilde{E}$, and its gradient with respect to the data $\partial E/\partial \boldsymbol{x}$, the force, are too small. Positive values of $d$ imply, by the same logic, that the force is too large. This explains the correlation between the force magnitude and $d$. Likewise, gradient descent will increase negative $d$ and decrease positive $d$, or equivalently increase forces that are too small and decrease forces that are too large. This explains the negative cross-correlation at short lags.

The difference in expected energies therefore corresponds to the change made to the overall magnitude of the energy at that point in training. Large fluctuations in the magnitude updates indicate instability and are undesirable.

### A.4 The temperature parameter

In the limit as the step size $\epsilon$ approaches zero, the discretized Langevin dynamics

$$\hat{\boldsymbol{X}}_{l+1} = \hat{\boldsymbol{X}}_l - \epsilon\frac{\partial E}{\partial \boldsymbol{x}}(\hat{\boldsymbol{X}}_l, \boldsymbol{\theta}) + \sqrt{2\epsilon}\hat{\boldsymbol{Z}}_l \tag{8}$$

has as its stationary distribution Eq. 1, which we repeat here for convenience:

$$\hat{p}(\boldsymbol{x}; \boldsymbol{\theta}) \propto \exp\{-E(\boldsymbol{x}, \boldsymbol{\theta})\}. \tag{1}$$

In practice when running Langevin dynamics, it is useful to be able to scale the gradient and noise terms independently: the latter must be small for stability, but the former must not be too small lest numerical

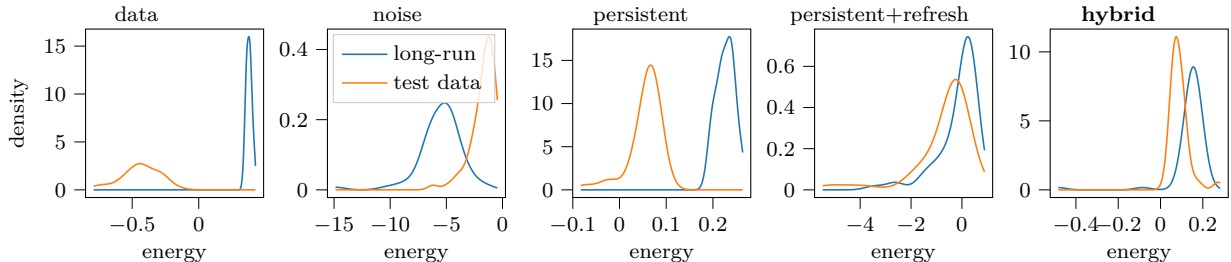

Figure 10: Distributions of energies assigned by models after training on the Oxford Flowers. Blue: long-run samples; orange: test (held-out) the Oxford Flowers samples. Plot titles specify the MCMC initialization scheme used during training.

precision of the gradient be lost. This is particularly true early in training, when the energy landscape produced by the neural network is very flat.

Including the parameter $T$ in Eq. 3 allows for independent scaling of the gradient and noise. Of course, this does change the stationary distribution; in particular, the Langevin update can be re-written as

$$\hat{\boldsymbol{X}}_{l+1} = \hat{\boldsymbol{X}}_l - \epsilon \frac{\partial E}{\partial \boldsymbol{x}}(\hat{\boldsymbol{X}}_l, \boldsymbol{\theta}) + \sqrt{2\epsilon T} \hat{\boldsymbol{Z}}_l$$

$$\eta := \epsilon T \implies \hat{\boldsymbol{X}}_{l+1} = \hat{\boldsymbol{X}}_l - \eta \frac{1}{T}\frac{\partial E}{\partial \boldsymbol{x}}(\hat{\boldsymbol{X}}_l, \boldsymbol{\theta}) + \sqrt{2\eta}\hat{\boldsymbol{Z}}_l, \tag{9}$$

and so the stationary distribution becomes

$$\hat{p}(\boldsymbol{x};\boldsymbol{\theta}) \propto \exp\{-E(\boldsymbol{x},\boldsymbol{\theta})/T\},$$

as can be seen by comparing Eqs. 8 and 1 with Eq. 9 (and grouping $1/T$ with the energy).

In the literature, $T$ is frequently omitted from the stationary distribution in order to reduce clutter, as we have in Eq. 1, and in the loss and its gradients, Eqs. 2, 5, and 6. Indeed, because *normalized* probabilities are rarely computable (or computed), $T$ is in some sense irrelevant to the reported results—it merely sets the "units" of the energy. However, in Fig. 2, we display the model densities themselves, and therefore have taken $T$ into account.

### A.5 The relative-entropy gradient for EBMs

Although it is well known, for completeness we include the derivation of the gradient of the relative entropy for energy-based models, here keeping the temperature parameter explicit:

$$\begin{aligned}
\frac{\mathrm{d}}{\mathrm{d}\boldsymbol{\theta}} \mathrm{D}_{\mathrm{KL}}\{p(\boldsymbol{X})\|\hat{p}(\boldsymbol{X};\boldsymbol{\theta})\} &= \mathbb{E}_{\boldsymbol{X}}\left[\frac{\mathrm{d}}{\mathrm{d}\boldsymbol{\theta}} \log \frac{p(\boldsymbol{X})}{\hat{p}(\boldsymbol{X};\boldsymbol{\theta})}\right] \\
&= \mathbb{E}_{\boldsymbol{X}}\left[\frac{\mathrm{d}}{\mathrm{d}\boldsymbol{\theta}} \log \frac{\int_{\hat{\boldsymbol{x}}} \exp\{-E(\hat{\boldsymbol{x}},\boldsymbol{\theta})/T\}\,\mathrm{d}\hat{\boldsymbol{x}}}{\exp\{-E(\boldsymbol{X},\boldsymbol{\theta})/T\}}\right] \\
&= \frac{\frac{\mathrm{d}}{\mathrm{d}\boldsymbol{\theta}}\int_{\hat{\boldsymbol{x}}} \exp\{-E(\hat{\boldsymbol{x}},\boldsymbol{\theta})/T\}\,\mathrm{d}\hat{\boldsymbol{x}}}{\int_{\hat{\boldsymbol{x}}} \exp\{-E(\hat{\boldsymbol{x}},\boldsymbol{\theta})/T\}\,\mathrm{d}\hat{\boldsymbol{x}}} + \mathbb{E}_{\boldsymbol{X}}\left[\frac{1}{T}\frac{\partial E}{\partial \boldsymbol{\theta}}(\boldsymbol{X},\boldsymbol{\theta})\right] \\
&= \frac{1}{T}\left(\mathbb{E}_{\boldsymbol{X}}\left[\frac{\partial E}{\partial \boldsymbol{\theta}}(\boldsymbol{X},\boldsymbol{\theta})\right] - \mathbb{E}_{\hat{\boldsymbol{X}}}\left[\frac{\partial E}{\partial \boldsymbol{\theta}}(\hat{\boldsymbol{X}},\boldsymbol{\theta})\right]\right).
\end{aligned} \tag{10}$$

Ignoring the temperature parameter (or setting it to unity) produces Eq. 2 of the main text.

