# OpenReview forum: "Revisiting Contrastive Divergence for Density Estimation and Sample Generation"
_TMLR — Accepted by TMLR_

### Review · Reviewer_vVbU · 2025-07-21

**Summary Of Contributions:**

The authors address a challenge with Energy Based Models that it is hard to have 1) good density estimation; 2) generating new samples from noise, and 3) efficient training (small number of Langevin steps) at the same time.

The first approach to remedy is by using persistent Markov chains where the gradient of the energy is estimated by an average of samples from persistent Markov chains.

The issue with the first approach is not allowing to sample freely from noise. To address that issue the authors revisit contrastive divergence methods.

The result is a loss function which has two components, which can be interpreted as the (short-run MCMC) + lambda * (contrastive-divergence gradient). The first term has initialized the Markov Chains at noise, and the second term has initialized the Markov chains at the data manifold.

The authors demonstrate that this hybrid loss works the best in both estimating the density and sampling from noise freely in a synthetic dataset of concentric circles.

An interesting empirical finding of this work is that the two loss terms do not conflict with each other, which is inline with the visualized hypothesis of the authors in Figure 1.

The authors demonstrate the usefulness of the hybrid approach on the Oxford flowers and CIFAR-10 datasets, as well as testing out of distribution energies.

The authors dig deeper to demonstrate that he hybrid approach yields the best SNR with training iterations.

**Audience:**

Yes

**Claims And Evidence:**

Yes

**Requested Changes:**

* How exactly do you compute the AUROC in Table 2? E.g. how do you measure exactly the distinguishing between the the energies from the in-distribution and the out-of-distribution datasets? Do you train a separate classifier? Can you clarify this in the main text?

**Strengths And Weaknesses:**

Strengths:

1. An excellent championing of contrastive divergence back to EBM. The authors defend the use of the hybrid approach, and explain why CD has been overlooked by the EBM literature. The hybrid approach is quite effective since the lambda=25 is in general transferrable, e.g. turning is not too hard.

2. Addition of a regularization term based on contrastive divergence is quite effective. It allows for density estimation (coming from the regularization term and good generation of samples from noise). I do appreciate the deep dives, especially the SNR computation.

3. The presentation of this work is clear and effective. I believe the audience will appreciate the insights.

Weaknesses:

1. I have a small question about the OOD detection, please see it below.

---

> ### Author Response · Authors · 2025-08-28
>
> We thank the reviewer for the careful reading of the manuscript, and are happy that he or she shares our enthusiasm for this case for contrastive divergence.  (We hope the reviewer will recommend this paper for conference presentation as well, where the results can reach a wider audience.)
>
> > How exactly do you compute the AUROC in Table 2?
>
> We have now added a brief description to the Methods, under "Out-of-distribution detection."  We reproduce it here for convenience:
>
> > We follow the standard procedure \citep{Hendrycks2017} to determine how well the model separates the probability densities of two datasets, in this case CIFAR-10 (on which EBM was trained) and SVHN.  In particular, we first calculate the energies for all images CIFAR-10 and SVHN.  We then calculate the fractions of true and false positives for every choice of an energy threshold putatively separating the two datasets.  Finally, we report the area under the graph of true positives vs. true negatives, i.e. the area under the receiver-operator characteristic (AUROC).

---

### Review · Reviewer_rfCw · 2025-08-08

**Summary Of Contributions:**

Authors tackles the problem of sampling from energy-based models.
Authors propose to combine contrastive-divergence and maximum likelihood.

**Audience:**

Yes

**Claims And Evidence:**

No

**Requested Changes:**

- Clarify the introduction: the problem setting and the challenges
- Provide the algorithm for training and inference
- Provide additional experiments (not necessarily SOTA) to back up the proposed loss

**Strengths And Weaknesses:**

Strengths:
- Authors tackle a hard and interesting problem
- I really like the emphasis of trying to bring an algorithm that converges with a practical number of sampling steps

Weaknesses:
- Clarity
- Empirical validation of the proposed loss

I detailed the 2 weaknesses below.

Disclaimer: I am not an expert in energy-based models

**Clarity**: I struggled to understand section 1 after equation 3.
- What does it mean to be a 'good generators without becoming good density estimators'?. In particular, experiments report the FID, which evaluates the generation quality.
- Can you clarify the following statement?  'And yet, contrary to the orthodox theory just reviewed here (and underlying the papers just cited), it seems that none of these EBMs has learned the data distribution (Nijkamp et al., 2019; 2020). More precisely, the distribution generated by the sampling process in Eq. 3 does resemble the data distribution, even though Eq. 1 does not'
    - What is exactly the setting here?
        - Do you have a 'reward' function you try to sample from?
        - You have a finite dataset and want to estimate the underlying probability distribution? With the specific shape of an energy function?
- ' In an important pair of papers, Nijkamp and colleagues showed that for the parameter choices typically employed in EBM training, in particular for any number of Langevin steps L less than ∼10,000 (L = 100 is typical), the Markov chain does not make it out of the burn-in stage. Consequently, the training
sculpts this energy landscape instead of that of the model'
Can you give more details on this? Especially, can you explain more the last sentence? I do not understand the causality link between the 2 sentences.
- ' “persistent initialization” of the Markov chain, i.e. initializing X̂ 1 at training iteration i with X̂ L from training iteration i − 1. (X̂ 1 at training
iteration i = 1 is drawn from a noise distribution.)  data dependent upon' I did not understand what is the 'persistence initialization'
- 'training procedure with L ∼ 100' I am confused here. From what I understand, L is the number of steps in the Langevin steps at inference. Why do you keep referring to L for the training procedure? Do authors do simulation-based training? Maybe if would help to write done the algorithm used for training and sampling.
- 'Short-run MCMC evidently sculpts a path from noise manifold to data manifold' Why is this 'evident'?
- ´The problem of “non-convergent MCMC” (Nijkamp et al., 2020) has its roots in the slow rate of convergence of Langevin dynamics for non-convex functions.´
Is this also something you observed in practice? That can be seen on the figures in the experiments.
- I am not sure if I understood the take-away of Figure 1.
- Overall the paper heavily relies on Nijkamp 2020, maybe it could be worth to provide extensive reminder at the beginning of the manuscript.

Nijkamp, E., Hill, M., Han, T., Zhu, S.C. and Wu, Y.N., 2020, April. On the anatomy of mcmc-based maximum likelihood learning of energy-based models. In Proceedings of the AAAI Conference on Artificial Intelligence (Vol. 34, No. 04, pp. 5272-5280).

**Experiments**:
- 'In contrast, models trained with noise-initialized Markov chains (second column) learn to generate from the data distribution (top row), but do not actually learn the correct energy (bottom row), as noted by Nijkamp and colleagues.'
- Can you give more details on the plots in Figure 5: what d they represent, and what is the takeaway from this Figure?
- Qualitative experiments are amazing to grab intuition (Figure 3 and 4 are very helpful). The only quantitative experiment I saw Figure 5 and Table 1 (+ the ablation study in Table 2.), that provides results for CIFAR-10.
Would it be possible to provide a wider range of experiments to support the claim the proposed loss is better/interesting.

Typos:
- page 11, durng >> during

---

> ### Author Response · Authors · 2025-08-28
> **recap of Nijkamp's results and EBMs**
>
> > Overall the paper heavily relies on Nijkamp 2020, maybe it could be worth to provide extensive reminder at the beginning of the manuscript.
>
> We take this to be the major issue underlying most of the reviewer's questions.  We provide here a longer, more detailed recapitulation of those results.  We spell the rationale out here very---perhaps excessively---explicitly because the reviewer describes him- or herself as not being an expert in EBMs.  We would like to reach such readers with our manuscript.  On the other hand, we expect most readers to be well-versed in EBMs.  Therefore we have foreborne adding this rationale to the MS, although we are open to doing so.  Perhaps some version of this material should be added as an appendix?
>
>
> The basic rationale behind energy-based modeling (as well as the more general task of fitting distributions to data) is that a model that can assign the correct probabilities to (held-out) data must have learned the structure underlying those data.  For example, if the model "knows" which configurations of pixels are improbable and which probable, then it "knows" what images look like.
>
> We can then use the knowledge of structure that is embedded in such models for various downstream tasks.  For example, we can ask how similar a new image is to the training set (e.g., "Is this image digit-like?" for an EBM trained on MNIST, or the OOD detection of our Table 2).  Or we can use the features extracted by the model as input to a classifier ("Is this a photo of a bird?").  Or again we might be able to generate new samples from the model that resemble the training distribution but are not identical to any training examples.
>
> We summarize the last point as: if an EBM assigns the correct probability to images, we might be able to generate good images from it.  What Nijkamp and colleagues showed, which was surprising at the time, is that the converse does not hold:  We might be able to generate good images from an EBM, even though it does *not* assign the correct probabilities to images.  They showed that such EBMs actually assign higher probabilities to highly saturated versions of the training images.  More precisely, the stationary distribution of an MCMC sampler applied to such an EBM contains only saturated images.
>
> The somewhat surprising assignment of low probability to the very data the model generates (under short-run LD) can be understood in part by recognizing (as Nijkamp and colleagues did) that in the standard recipe for training EBMs, the Langevin dynamics are not run long enough, either during training or during testing, to exit the burn-in stage of the Markov chain:  The resulting samples are *not* from the stationary distribution of the EBM---that is, they are not samples to which the EBM assigns high probability.  (Recall that in Markov chain Monte Carlo, one constructs a transition operator with the property that repeated application ultimately generates samples from a stationary distribution that is independent of the initial input to the operator.  Early samples in the chain are typically *not* independent of the initial sample; this part of the sampling process is known as the "burn-in" period.)
>
> Thus, during training, using these "short-run" samples to approximate an expectation under the model distribution (see Eq. 2 of our MS) yields the wrong gradient, and therefore the wrong weight updates.  This explains why the resulting model assigns the wrong probability to training data.  But it still leaves obscure how the model generates such good images.  We explained this, following [Xiao et al. 2021b], by interpreting the training procedure as GAN training (in the section "Noise-initialized LD for generation").

---

> > ### Author Response · Authors · 2025-08-28
> > **reviewer questions**
> >
> > > What does it mean to be a 'good generators without becoming good density estimators'?. In particular, experiments report the FID, which evaluates the generation quality.
> >
> > Some of our experiments test the quality of generation, and some test the quality of density estimation.  To determine the quality of generation, we examine images made by the generator, namely, short-run Langevin dynamics carried out on the EBM.  We use both FID scores (Table 1) and "the eye test" (the top rows of Figs. 3 and 4).
> >
> > Prior to Nijkamp et alia's papers, this would also have been taken as evidence of good density estimation, but now we know that these generated samples might not actually represent high probability data under the EBM (see above).  So to determine the quality of density estimation, we need another probe.  We use several.
> >
> > First, an EBM that is a good density estimator will assign low energy (high probability) to all images from the target dataset, even images held out from training.  We cannot assay this directly because we cannot say a priori how low is "low."  Instead, then, we initialize the LD at such images.  If they are highly probable under the model, then the LD is unlikely to move away from them toward lower-quality images; that is, they should be within the stationary distribution of the Markov chain.  (This nicely sidesteps the question of how many LD steps to take from noise to reach the stationary distribution, and can also be carried out in fewer steps.)  We find that LD drifts toward worse images (only) for EBMs trained with noise-initialized chains (bottom row of Figs. 3 and 4).
> >
> > However, although necessary for being a good density estimator, passing this test is not sufficient, since the EBM might also assign low energies (high probabilities) to other, low-quality images, or have poor mode coverage.  To test this, we draw samples that we can be confident are highly probable under the EBM, namely, samples from very long-run (100,000-step) Markov chains.  By 100k steps, LD reaches its stationary distribution even for EBMs trained on complex data sets.  Since the stationary distrbution under Langevin dynamics corresponds to the probability density encoded by the EBM (with the caveats about the Metropolis adjustment discussed in the text), samples from the stationary distribution tell us about the quality of density estimation.  We find that such samples from the hybrid-trained EBM, but from no others, resemble the data and have good mode coverage (middle rows, Figs. 3 and 4).
> >
> > Third, to confirm this intuition, we can directly compute the EBM energies of long run samples (i.e., what the EBM assigns high probability to) and of samples from the data distribution (images from CIFAR-10).  We see (Fig. 5) that both types of persistently-trained models assign similar energy to their own bad samples as to the samples from the data distribution, which confirms that, although these models "like" good samples, they also "like" bad samples---so they are not good density estimators.
> >
> > Fourth, and along similar lines, we expect a good density estimator for CIFAR-10 to assign very different energies to images that don't resemble CIFAR-10---e.g., from the SVHN dataset.  If that is the case, then the distributions of energies from CIFAR-10 and SVHN should be highly separable.  We measure this with the standard tool from the literature, area under the receiver-operator characteristic (AUROC).  Here we find that EBMs trained with hybrid-initialized chains and EBMs trained with data-initialized chains (CD) yield the best AUROC.
> >
> > > Can you clarify the following statement?  'And yet, contrary to the orthodox theory....'
> >
> > We believe we have clarified this with the meticulous description above.
> >
> > > Do you have a 'reward' function you try to sample from?  You have a finite dataset and want to estimate the underlying probability distribution?  With the specific shape of an energy function?
> >
> > We follow the standard procedure for fitting parameterized probability distributions (Eq. 1) to data:  We descend the gradient of the relative entropy between the data distribution and the model distribution (Eq. 2).  We have a finite set of samples from the data distribution (e.g., CIFAR-10 contains 60,000 images).  The shape of the energy function is extremely flexible, since it is given by a multi-layer neural network.  (We're not sure what the reviewer means by a reward function that we try to sample from.)

---

> > > ### Author Response · Authors · 2025-08-28
> > > **more reviewer questions**
> > >
> > > > I did not understand what is the 'persistence initialization'
> > >
> > > Persistent initialization is no more (or less) than the sentence the reviewer has quoted here.  But we also describe it at greater length below in the section "Persistent Markov chains?"  Various flavors are described in the references cited therein (Du & Mordatch 2019, Du et al. 2020, Tieleman 2008).
> > >
> > > > From what I understand, L is the number of steps in the Langevin [dynamics] at inference.  Why do you keep referring to L for the training procedure?  Do authors do simulation-based training?  Maybe if would help to write done the algorithm used for training and sampling.
> > >
> > > To train the model, it is necessary to draw samples, so as to approximate the second expectation in the gradient, Eq. 2.  We say this right after that equation ("The dominant approach to approximating Eq. 2 is therefore....").  If the goal is to generate samples, then at test time (we would not call this "inference"), one also runs Langevin dynamics.
> > >
> > > To clear up any remaining confusion, we have added pseudocode for training and testing to the manuscript.
> > >
> > > > Short-run MCMC evidently sculpts a path from noise manifold to data manifold' Why is this 'evident'?
> > >
> > > See "recap of Nijkamp's results and EBMs" above.
> > >
> > > > 'The problem of "non-convergent MCMC" (Nijkamp et al., 2020) has its roots in the slow rate of convergence of Langevin dynamics for non-convex functions.'  Is this also something you observed in practice? That can be seen on the figures in the experiments.
> > >
> > > Yes, this can be seen (e.g.) in the images generated from EBMs trained with noise-initialized or persistently initialized chains, e.g. Figs. 3 and 4, second and third columns.  Short-run (500-step) samples (top row) do not resemble long-run (100,000-step) samples (middle row).  That means that the Markov chain still has not converged after 500 steps.  In fact, Nijkamp et al. estimate that on the order of 10,000 steps are needed to reach convergence; please see their papers for more visual examples.
> > >
> > > > I am not sure if I understood the take-away of Figure 1.
> > >
> > > We hope this is clearer in light of our answers above.  Briefly, the figure shows what we believe to be the effect of training with short-run LD initialized at noise, namely that the energy will run downhill (black arrows) from the noise vectors (red) to the true manifold of images (blue); as well as the effect of training with contrastive divergence (i.e., short-run LD initialized at data), namely that moving away from the data manifold in *any* direction (green arrows) goes uphill.  But this would have been clearer if we had put green arrows on the inside of the data manifold as well as the outside, which we have now done.
> > >
> > > > Can you give more details on the plots in Figure 5: what do they represent, and what is the takeaway from this Figure?
> > >
> > > Please see our response to the question "What does it mean to be a 'good generators without becoming good density estimators'?"

---

> > > > ### Author Response · Authors · 2025-08-28
> > > > **experiments**
> > > >
> > > > > The only quantitative experiment I saw Figure 5 and Table 1 (+ the ablation study in Table 2.), that provides results for CIFAR-10. Would it be possible to provide a wider range of experiments to support the claim the proposed loss is better/interesting.
> > > >
> > > > Table 2 is not an ablation study but out-of-distribution detection; please see our response to the question "What does it mean to be a 'good generators without becoming good density estimators'?" for what these results say about density estimation.
> > > >
> > > > We also quantify the variance and signal-to-noise ratios of the various algorithms in Figure 7.  We believe a key reason that persistent methods, which are superficially similar to ours, do not work as well is that they have higher variance (for the reasons given in the text).
> > > >
> > > > We will shortly also post the results of two more quantitative experiments:
> > > >
> > > >  - FID for long-run chains
> > > >  - The distributions of energies, as in Figure 5, but for Oxford Flowers
> > > >
> > > > We have also added a figure showing that the EBM trained with the hybrid method is not merely memorizing the training data: we show that samples from an EBM trained on CIFAR-10 do not closely resemble any of their three (pixelwise-)nearest neighbors in the training set.

---

> > > > > ### Author Response · Authors · 2025-09-02
> > > > > **additional experiments**
> > > > >
> > > > > We have added three additional experiments (alluded to above):
> > > > >
> > > > > 1. FID for long-run chains for EBMs trained with the hybrid loss on CIFAR-10.  We draw the following conclusions (in the text):
> > > > >
> > > > > > Note that if we continue to run the Langevin dynamics in the hybrid-trained EBMs for a very long time ($L_\text{test}$ = 100, 000), the FID score does not much change (middle row of Table 1, FID = 35.3). This is consistent with the observation that the short- and long-run samples from this model look qualitatively similar to each other (Fig. 3, last column, top and middle rows), and corroborates the conclusion that short-run LD convergences to the stationary distribution in EBMs trained with the hybrid loss, whereas it does not under the noise-initialized loss and its variants.
> > > > >
> > > > > 2. The distributions of energies for as in Fig. 5, but for Oxford Flowers.  This is Fig. 10 in the manuscript (in the Appendix).  We observe the same general pattern, which we describe informally here:
> > > > >  - CD training (i.e., data-initialized LD) yields a model that "prefers" true flowers (test data) to its own long-run data---because the model never converges when initialized at noise.
> > > > >  - Training with noise-initialized LD yields a model that prefers its own (saturated), long-run samples to true flowers---because sampling during training never exits the burn-in stage.
> > > > >  - Training with persistent Markov chains yields a model that has prefers true flowers to its own long-run data---so the model does not converge.
> > > > >  - Training with persistent Markov chains, periodically restarted at noise ("refreshed"), yields a model that has about equal preference for its own long-run samples as for true flowers.  This is not as damning as in the case of CIFAR-10, because the long-run samples are acceptable--although not very diverse.
> > > > >  - Training with the hybrid loss yields a model with a similar (but slightly greater) preference for true flowers and its own long-run samples, which shows that model has the correct stationary distribution.
> > > > >
> > > > > 3. Sample diversity.  We show that EBMs trained with the persistent+refresh scheme do not generate diverse samples under their preferred generation scheme, namely, initialization at a sample bank.  In contrast, EBMs trained with the hybrid loss do generate diverse samples under their preferred generation scheme, initialization at noise.  We have a new section (4.2.4), figure (now Fig. 7), and discussion of this, which we paste here for convenience:
> > > > >
> > > > > > ...EBMs trained with persistent Markov chains generate higher-quality examples at test-time when initialized at samples
> > > > > from a bank (accumulated during training) rather than at noise samples (recall Table 1). But the price to be paid is that, at least without more “tricks,” a generated sample will frequently be very similar to the sample from the bank at which the LD was initialized.  We show this in Fig. 7. Fig. 7A shows a sample bank accumulated during training of an EBM on the Oxford Flowers, and Fig. 7B shows the samples generated from the trained EBM after L = 500 steps of LD, starting form the sample
> > > > > bank. The resemblance between Fig. 7A,B is very strong; we might even say that the “generation” process is mostly just copying samples out of the bank.
> > > > >
> > > > > > In contrast, EBMs trained under the hybrid loss generate novel samples. Fig. 7C shows samples from such
> > > > > an EBM in the first column. The next three columns show their (pixelwise) nearest neighbors in the training
> > > > > set. Evidently, the generated images do not closely resemble the training data, which shows that the EBM
> > > > > has learned the structure of (CIFAR-10) images rather than merely memorizing examples.

---

### Review · Reviewer_grgR · 2025-08-10

**Summary Of Contributions:**

The main contributions of this paper are: 1. The author(s) compared several different training schemes of image generation, discussing their strengths and weaknesses and performed experiments to validate them; 2. The author(s) proposes a method which combines both data- and noise-initialized schemes; 3. The author(s) show by experiments that this new method performs well on both synthetic and natural datasets.

**Audience:**

Yes

**Broader Impact Concerns:**

No.

**Claims And Evidence:**

Yes

**Requested Changes:**

Some details in this paper are not made clear enough. See below.

1. The variable $T$ in eq (3) is not defined.
2. There are many things like “$L \sim$ 100/500/etc.” The author(s) should make clear how the “$\sim$” is defined. Since we are dealing with finite numbers here, for example $L \sim 100$ should not mean "$100 c_1 \le L \le 100 c_2$ for some constants $c_1$, $c_2$.
3. In the experiment for synthetic dataset, the rings are in 2D plane, the meaning of “xyz-plane”, especially the z-direction, in the first paragraph of “Section 3, Synthetic dataset” should be made clear.
4. In Figures 2,3,4, I suggest changing the title “Initialization during training” to something like “Results of different training schemes”.
5. There are many “$\hat{X}_{subscript}$” in the paper. It would be helpful to mention their meanings. For example, do the $\hat{X}_l$ in eq (3) the same as the samples $\hat{X}_n$ in Section 4.2.4?

**Strengths And Weaknesses:**

Strengths: I am not familiar with image generation, but I learned much from this work. I think the paper is well-structured, mostly self-contained, and provides useful, sufficient information to the audience. While the idea behind their hybrid loss and training scheme sounds natural (combining two to achieve more), the results are solid on both synthetic and natural datasets.

Weaknesses: I do not see this paper has significant weaknesses. But I do have some questions, mostly out of curiosity for this work.

1. How is the complexity of the training scheme proposed by the author(s), compared to the other training schemes (noise, data, persistent, etc.). It would be helpful to either discuss it or illustrate it in the paper.

2. Why the $\lambda = 25$ gives the best performance? How does the best value of $\lambda$ depend on the training/testing set? It would be good to discuss it, even if it is just for the experiments done in this paper.

3. I wonder if there is any theoretical results regarding the better behavior of the training scheme proposed by the author(s).

---

> ### Author Response · Authors · 2025-08-28
>
> We thank the reviewer for the careful reading of the ms.---and the positive appraisal.  (We hope the reviewer will recommend this paper for conference presentation as well!)
>
> > How is the complexity of the training scheme proposed by the author(s), compared to the other training schemes (noise, data, persistent, etc.). It would be helpful to either discuss it or illustrate it in the paper.
>
> The algorithm (now made explicit in Algorithm 1 in the manuscript) requires a second run of LD and a second gradient evaluation (forward and backward pass through the model), so it incurs roughly a factor of 2 increase in computational costs over the standard, noise-initialized training.
>
> We have now put the algorithm (in pseudo-code) in the manuscript, so perhaps this will be easier to see.
>
> > Why the $\lambda = 25$ gives the best performance? How does the best value of $\lambda$ depend on the training/testing set? It would be good to discuss it, even if it is just for the experiments done in this paper.
>
> We have now included FID scores for models trained with the hybrid loss with $\lambda=10$ and $\lambda=50$ (as for OOD detection).  We provide some context for this result in a (new) final paragraph in section 4.2.2., which we repeat here for convenience:
>
> > That OOD detection improves monotonically with the weight on the data-
> initialized loss ($\lambda$), but FID does not, is intuitive and related to the fact that in general we require $\lambda \gg 1$, i.e. that the data-initialized loss be weighted much more highly than the noise-initialized loss. The EBM needs to learn a more detailed model of the data than of the noise distribution, or of the path from noise to data, since it does not matter much how the sampler arrives at the data manifold, as long as it does; whereas generating high-quality images or assigning the right probabilities to data requires many more bits.
>
> We also explicitly summarize the trend in the Discussion:
>
> > Increasing the weight on the data-initialized chain beyond this point [$\lambda=25$] (i.e., making training more CD-like) improves out-of-distribution detection (Table 2), but makes image generation worse (Table 1).
>
> We did not exhaustively optimize $\lambda$ for different datasets---but we see it as a feature rather than a bug that a single value ($\lambda=25$) works well for all experiments.
>
> > I wonder if there is any theoretical results regarding the better behavior of the training scheme proposed by the author(s).
>
> We do not immediately see a way to apply existing theory about convergence in LD (e.g., in the technical report from Xiang Cheng at al. that we cite) to our problem setup, although we do believe this would be useful.  For example, for CIFAR-10 and Oxford Flowers, we find that a hybrid-trained model needs approximately 500 Langevin steps to reach its stationary distribution from noise.  But models trained only with noise-intilialization *also* need about 500 steps to reach similarly good-looking images---yet these are far from its stationary distribution.  This strongly suggests that convergence properties are determined by non-equilibrium phenomena not closely related to the stationary distribution of the Markov chain.  This is a challenging context for theory.
>
> > 1. The variable $T$ in eq(3) is not defined
>
> Thank you for pointing this out.  Briefly, a temperature parameter $T$ is typically included in Langevin dynamics for better numerical performance, and the stationary distribution and loss modified accordingly.  This is seldom made explicit in the literature on EBMs, but is universal in practice (in the code).
>
> In the interest of readability but to retain consistency with the other equations, we have removed $T$ from Eq. 3.  Then we have added a section the Appendix in which $T$ and its roles in the Langevin dynamics (Eq. 3), the stationary distribution (Eq. 1), and the loss and its gradients (Eqs. 2, 5, and 6) are made explicit.
>
> > 2. There are many things like $L \sim 100$
>
> We meant this in the sense of "approximately," but seeing the ambiguity, we have now replaced $\sim$ with $\approx$ or the word "approximately."
>
> > 3. ...the meaning of “xyz-plane”, especially the z-direction...should be made clear
>
> We have now clarified this section of the text (in the subsection "Synthetic dataset" under the Methods).
>
> > 4. I suggest changing the title “Initialization during training”....
>
> We intended this line to state what the column headings are specifying (just as "test-time initialization" state what the row headings are specifying).  We have now made this more explicit by writing, "columns: initialization during training," and likewise for the rows.

---

> ### Author Response · Authors · 2025-08-28
>
> 5. We use $L$ for the number of Langevin steps, with $l$ for the corresponding index; and $N$ for the batch size, with $n$ for the corresponding index.  (We also reserve $i$ for the iteration of gradient descent.)  We have now defined $N$ at its introduction in S4.2.4 (and left $n$ to be defined by context).  We also can see how using subscripts to indicate both batch index (if the letter is $n$) and Langevin step (if the letter is $l$) is confusing, so we have pushed the former into superscript. (We defined $L$ where it is first introduced, so we believe this clears up any remaining ambiguity.)

---

### Author Response · Authors · 2025-09-02
**updated manuscript**

In response to reviewer questions and concerns, we have made many revisions to the MS, detailed above.  We now await the reviewers' replies.

---

> ### Author Response · Authors · 2025-09-26
> **final recommendations?**
>
> It has been several weeks since the discussion period began (Aug. 8?).  Is there any update on the decision process?

---

### Decision · Action_Editor_XjJJ · 2025-09-26

**Recommendation:** Accept as is

**Audience:**

Yes

**Audience Explanation:**

Yes, TMLR's audience would likely find this work of interest. Energy-Based Models represent an active area of machine learning research, and the fundamental challenge addressed—achieving both good density estimation and sample generation with practical computational requirements—is a core problem that has hindered EBM adoption. The paper's revival of contrastive divergence within a hybrid training framework offers a novel solution that could influence how practitioners approach EBM training. Additionally, the work builds meaningfully on influential recent findings by Nijkamp et al. about non-convergent MCMC in EBMs, making it relevant to researchers working on generative modeling, MCMC methods, and energy-based approaches.

**Claims And Evidence:**

Yes

**Claims Explanation:**

The claims in this submission are generally supported by adequate evidence. The authors provide convincing empirical validation through visual assessment of generated samples, quantitative metrics (FID scores), density estimation quality tests, and out-of-distribution detection performance. The synthetic dataset experiments effectively demonstrate simultaneous good generation and density estimation, while CIFAR-10 and Oxford Flowers results show practical applicability.